# CPT: CONSISTENT PROXY TUNING FOR BLACK-BOX MODELS

## ABSTRACT

Black-box tuning has attracted recent attention due to that the structure or inner parameters of advanced proprietary models are not accessible. Proxy-tuning (Liu et al., 2024) provides a test-time output adjustment for tuning black-box language models. It applies the difference of the output logits before and after tuning a smaller white-box "proxy" model to improve the black-box model. However, this technique serves only as a decoding-time algorithm, leading to an inconsistency between training and testing which potentially limits overall performance. To address this problem, we introduce **C**onsistent **P**roxy **T**uning (**CPT**), a simple yet effective black-box tuning method. Different from Proxy-tuning, CPT additionally exploits the frozen large black-box model and another frozen small white-box model, ensuring consistency between training-stage optimization objective and test-time proxies. This consistency benefits Proxy-tuning and enhances model performance. Note that our method focuses solely on logit-level computation, which makes it model-agnostic and applicable to any task involving logit classification. Extensive experimental results demonstrate the superiority of our CPT in both black-box tuning of Large-Language Models (LLMs) and Vision-Language Models (VLMs) across various datasets.

## 1 INTRODUCTION

Although large-scale pretrained models have demonstrated strong generalization capabilities, they can perform better on specific downstream tasks by fine-tuning. Several parameter-efficient fine-tuning techniques have been developed to fine-tune Large Language Models (LLMs), such as soft prompt tuning (Lester et al., 2021), adapters (Houlsby et al., 2019), Low-Rank Adaption (LoRA) (Hu et al., 2021) and sparse tuning (Zaken et al., 2021). Similar approaches are also applied to fine-tune the pretrained Vision-Language Models (VLMs), including text/visual prompt tuning (Zhou et al., 2022b;a; Bahng et al., 2022), adapter-based tuning (Zhang et al., 2022; Gao et al., 2024), etc. Notice that these fine-tuning methods are usually under the strong assumption that the model architectures are known and model parameters are accessible. However, for privacy or commercial reasons, some advanced proprietary models are closed-source (*i.e.*, black-box models). For instance, users of GPT-4 (Achiam et al., 2023) can only interact with the model through a controlled interface and cannot access the model's parameters or intermediate embeddings. Therefore, these white-box optimization methods are infeasible for tuning the black-box models. Some methods do focus on tuning the black-box models. For example, BBT (Sun et al., 2022b) and BBT-v2 (Sun et al., 2022a) employs gradient-free strategies for fine-tuning of black-box LLMs, while CBBT (Guo et al., 2023) and LFA (Ouali et al., 2023) fine-tune VLMs by adaptive or aligned features. Nevertheless, all these methods require access to features within the model, which is not applicable for more strict black-box scenarios.

Recently, Proxy-tuning (Liu et al., 2024) improves the large black-box model with proxy strategy, *i.e.*, improves the large black-box model by tuned/untuned smaller white-box models. Specifically, during inference, the difference in logits between a tuned small model and an untuned small model is used as an offset and added to the output logits of the large black-box model to generate the final prediction. Proxy-tuning is more widely applicable and privacy-preserving compared to other existing methods in that it requires minimum access to black-box models—basic access to output logits will suffice. However, we note that there is an inconsistency between the optimization objective of Proxy-tuning (Liu et al., 2024) and the form of output ensemble during inference, *i.e.*, only a small white-box model alone is used for tuning while the ensemble of three models are used for

predictions. Such inconsistency may lead to sub-optimal solutions for the proxy tuning optimization objective, causing bottlenecks in model performance.

To *reconcile the inconsistency in Proxy-tuning* (Liu et al., 2024), in this paper we propose **C**onsistent **P**roxy **T**uning (CPT), a simple yet effective proxy-tuning method. Specifically, during training stage of the tunable small white-box model, CPT additionally incorporates the frozen large black-box model and another frozen small white-box model. Given a training sample, these three models first compute the logit scores for the input sample respectively. Then, the three sets of logits are ensembled by using the same logits calculation formula of a test-time proxy in Proxy-tuning (Liu et al., 2024). Finally, the tunable white-box model is optimized with the loss function computed with the ensemble logits and the ground truth. During inference stage, we follow Liu et al. (2024) to employ the proxy tuning for large black-box model. The whole pipeline of CPT is illustrated in Fig. 1. Compared to vanilla Proxy-tuning (Liu et al., 2024), our CPT ensures consistency between the optimization objective during the training of the small white-box model and the test-time inference with proxy. This benefits the Proxy-tuning process and enhances the performance of the model.

Note that our method focuses solely on logit-level computation, thus holding the potential of being a plug-and-play improvement for any black-box model fine-tuning tasks which involve logit-level classification. In this paper, we show the effectiveness of CPT by applying it to two representative black-box tuning tasks respectively: black-box tuning of Large-Language Models (LLMs) and black-box tuning of Vision-Language Models (VLMs). For black-box tuning of LLMs, we use a LLAMA2 (Touvron et al., 2023) model with a lightweight architecture (*e.g.*, LLAMA2-7B) to consistently proxy-tune the LLAMA2 model with heavier architecture (*e.g.*, LLAMA2-13B) on various downstream natural language processing tasks. Our CPT outperforms Proxy-tuning (Liu et al., 2024) by **2.20**% in terms of mean accuracy across seven natural language processing datasets. For black-box tuning of VLMs, we use a CLIP (Radford et al., 2021) with a lightweight image encoder (*e.g.*, ResNet-50 (He et al., 2016)) to consistently proxy-tune the CLIP model with a heavier image encoder (*e.g.*, ViT-B/16 (Dosovitskiy et al., 2020)) on image classification task. Our CPT outperforms Proxy-tuning (Liu et al., 2024) by **1.24**% in terms of mean accuracy across eight image classification datasets.

In a nutshell, the main contributions of this paper are summarized as follows:

1) We propose **C**onsistent **P**roxy **T**uning (CPT), a simple yet effective proxy-tuning method for black-box model optimization.

2) CPT introduces a frozen black-box large model and a frozen white-box small model into the training of another tunable white-box small model, which ensures that the optimization objectives during white-box training are consistent with the form of proxy-tuning during inference.

3) CPT can be widely applied to a variety of black-box model fine-tuning tasks. Extensive experiment results of the black-box tuning for VLMs and LLMs on various datasets demonstrate the effectiveness of our CPT.

## 2 RELATED WORK

**Efficient Fine-tuning.** Large pretrained models, which are extensively trained on vast datasets, demonstrate broad generalization capabilities across various tasks. To further improve the performance of these models on specific downstream tasks, efficiently fine-tuning methods have been proposed for large pretrain models. In the field of natural language processing, some approaches focus on designing lightweight components to fine-tune pretrained Large Language Models (LLMs). For example, soft prompt tuning (Lester et al., 2021) introduces continuous learnable prompts other than hard prompts. Adapter-based method (Houlsby et al., 2019) inserts learnable adapters into Transformer (Vaswani et al., 2017), thus transferring to downstream tasks while preserving pretrained knowledge. Low-Rank Adaptation (LoRA) (Hu et al., 2021) freezes the pretrained model weights and injects trainable rank decomposition matrices into each layer of the Transformer. BitFit (Zaken et al., 2021) only tunes the bias terms of the model.

Many other works also explore how to efficiently fine-tune pretrained VLMs (*e.g.*CLIP (Radford et al., 2021)). CoOp (Zhou et al., 2022b) designed learnable text prompts to better understand natural language context. Then CoCoOp (Zhou et al., 2022a) further uses images as conditions to constrain the optimization of text prompts. Visual prompting (Bahng et al., 2022) also shows

that visual prompting is particularly effective for CLIP. Some works (Zhang et al., 2022; Gao et al., 2024) adopts add adapters to the encoders of CLIP, thus to fit different tasks while preserving pretrain knowledge.

However, these above strategies require access to the model's internal parameters (white-box access), which is not feasible for many of today's sophisticated models. Such infeasibility calls for new paradigms in fine-tuning black-box pretrained models.

**Black-Box Tuning.**   Black-box large pretrained models require a special set of fine-tuning methods. For large language models, BBT  (Sun et al., 2022b) achieves gradient-free optimization by using covariance matrix adaptation evolution strategy (CMA-ES)  (Hansen et al., 2003).However, it requires permission from black-box model to use customized prompt embedding, which is not feasible with some popular language models *e.g.*, GPT-4  (Achiam et al., 2023). BBT-v2  (Sun et al., 2022a) injects learnable prompt into layers of the LLM, which is also not applicable for language model APIs. BDPL  (Diao et al., 2022) investigates the possibilities of using discrete prompt to help LLMs understand the task better. Proxy-tuning  (Liu et al., 2024) considers training smaller white-box models as proxy instead, and use the fine-tuned white-box experts to enhance black-box LLMs. This approach has shown both effectiveness and promise, but it overlooks the inconsistency between the training objective of small model and the joint test-time ensemble of large black-box model and smaller proxy models. Zhang et al. (2020) also uses small models to indirectly fine-tune large models, but they need access to the parameters of intermediate layers, which is not suitable for scenarios where the parameters of the model cannot be accessed.

For vision-language models, BlackVIP  (Oh et al., 2023) optimizes the coordinator which generates visual prompts by zeroth-order optimization. However, the improvement in performance is limited. Linear Feature Alignment (LFA)  (Ouali et al., 2023) optimizes a projection layer to enhance the alignment between pre-computed image features and class prototypes. CBBT  (Guo et al., 2023) optimizes textual prompt and feature output adaptation collaboratively. These two methods interact with the black-box models at a feature level and requires access to output features, which leaves a potential risk of being vulnerable to attacks *e.g.*membership inference attacks (MIA)  (Carlini et al., 2022). We focus on a more restrict black-model setting where only output logits other than output features of a model is accessible.

**Logits Arithmetic.**   Recently, some methods (Dou et al., 2019) have demonstrated the capability of logits ensembling from multiple models in enhancing model performance. For example,  Dou et al. (2019) assembles multiple logits from models pretrained on different domains to achieve domain adaptation. DExperts  (Liu et al., 2021) uses the difference in logits output between a toxic model and a non-toxic model to assist language models in language detoxification. This paper also explores the use of proxy (Liu et al., 2024) to "fine-tune" Large Language Models (LLMs) during the inference stage. Contrastive Decoding (CD) (Li et al., 2022) leverages the differences in log-likelihood between expert and amateur language models (LMs) of varying sizes by selecting tokens that maximize this discrepancy. Some subsequent studies have also explored the effects of ensembling output logits from different layers of models (Gera et al., 2023; Chuang et al., 2023) or the effects of output logits from varying inputs (Pei et al., 2023; Shi et al., 2023). This paper proposes a method that enhances a large black-box model using the output from a small white-box model and an untuned one. Unlike Proxy-tuning (Liu et al., 2024), which overlooks the consistency between proxy-independent training and proxy-dependent testing and results in suboptimal outcomes, our method employs ensembled output logits from both black-box and white-box models as optimization objectives. This approach ensures consistency in proxy techniques, thereby enhancing model performance.

## 3  PROPOSED METHOD

### 3.1  REVISITING PROXY-TUNING

Given a large black-box LLM $\mathcal{M}_l(\cdot; \boldsymbol{\theta}_l^p)$ with inaccessible pretrained parameters $\boldsymbol{\theta}_l^p$, we only assume access to the output logits across the entire output space. Since $\mathcal{M}_l$ is a black-box model, directly fine-tuning it on downstream datasets with methods such as full fine-tuning or LoRA (Hu et al., 2021) is not applicable, as these methods require access to the model parameters. To tackle this problem, the novel practice of tuning models by proxy  (Liu et al., 2024) improves a large black-box model $\mathcal{M}_l$ with proxy *i.e.*, smaller tuned white-box models. Specifically, during training stage,

Figure 1: Illustration the comparison of our **C**onsistent **P**roxy **T**uning (CPT) with vanilla Proxy-tuning (Liu et al., 2024). (a) and (b) respectively illustrate the training and inference stage of Proxy-tuning. Notice that their optimization objectives and the formula of the proxy during inference are inconsistent. In contrast, our CPT achieves consistency in these two aspects, as shown in (c). Especially, when $\alpha_{train} = 0$ and $\alpha_{test} = 1$, our CPT will degenerate into the "inconsistent" Proxy-tuning.

a small white-box model $\mathcal{M}_s(\cdot; \boldsymbol{\theta}_s^p)$ with pretrained parameters $\boldsymbol{\theta}_s^p$ is fine-tuned by downstream dataset $\mathcal{D}$ with supervised learning paradigm. Given an input $\mathbf{x}$ and corresponding ground truth $\mathbf{y}$, the model is fine-tuned with the optimization objective of

$$\boldsymbol{\theta}_s^t = \arg\min_{\boldsymbol{\theta}_s} \mathbb{E}_{(\mathbf{x},\mathbf{y})\sim\mathcal{D}}[\mathcal{L}(\mathcal{M}_s(\mathbf{x}; \boldsymbol{\theta}_s), \mathbf{y})], \tag{1}$$

where $\mathcal{M}_s(\mathbf{x}; \boldsymbol{\theta}_s)$ with parameters $\boldsymbol{\theta}_s$ denotes the output logits of model $\mathcal{M}_s$, $\boldsymbol{\theta}_s^t$ denotes the optimized parameters and $\mathcal{L}$ is the classification loss function, *e.g.* cross entropy loss. During the inference stage, a test data $\mathbf{x}$ is fed to $\mathcal{M}_s(\cdot; \boldsymbol{\theta}_s^t)$, $\mathcal{M}_s(\cdot; \boldsymbol{\theta}_s^p)$ and $\mathcal{M}_l(\cdot; \boldsymbol{\theta}_l^p)$ to obtain output scores $\mathbf{z}_{\mathcal{M}_s^t}$, $\mathbf{z}_{\mathcal{M}_s^p}$ and $\mathbf{z}_{\mathcal{M}_l^p}$, respectively. Then, the final prediction probability of proxy-tuned models on input $\mathbf{x}$ can be formally expressed as:

$$p(\mathbf{x}) = \mathbf{z}_{\mathcal{M}_s^t} + (\mathbf{z}_{\mathcal{M}_l^p} - \mathbf{z}_{\mathcal{M}_s^p}). \tag{2}$$

Eqn. 1 indicates that only the output of the small model $\mathcal{M}_s$ is involved in optimization during training stage. However, during the inference stage, the final prediction score is calculated by ensembling the outputs from all three models, as shown in Eqn. 2. This inconsistency between training and inference (Fig. 1 (a) and Fig. 1 (b)) limits the training process to only finding a sub-optimal solution for the proxy-tuning model.

### 3.2 CONSISTENT PROXY TUNING (CPT)

In this paper, we aim to bridge the inconsistency between the use of test-time proxies and the separate training process small white-box model. To this end, we propose **C**onsistent **P**roxy **T**uning (CPT) method, which is illustrated in Fig. 1 (c). In contrast to the vanilla Proxy-tuning (Liu et al., 2024), CPT additionally incorporates frozen $\mathcal{M}_s(\cdot; \boldsymbol{\theta}_s^p)$ and $\mathcal{M}_l(\cdot; \boldsymbol{\theta}_l^p)$ into the fine-tuning process of the small white-box model, and the optimization objective is improved to compute the loss function based on the ensemble of the outputs from the three models and the ground truth. Formally, the optimization objective is modified as follows:

$$\boldsymbol{\theta}_s^t = \arg\min_{\boldsymbol{\theta}_s} \mathbb{E}_{(\mathbf{x},\mathbf{y})\sim\mathcal{D}}[\mathcal{L}(\mathcal{M}_s(\mathbf{x}; \boldsymbol{\theta}_s) + \alpha_{train}(\mathcal{M}_l(\mathbf{x}; \boldsymbol{\theta}_l^p) - \mathcal{M}_s(\mathbf{x}; \boldsymbol{\theta}_s^p)), \mathbf{y})], \tag{3}$$

where $\alpha_{train}$ is the coefficient that controls the impact of the offset obtained from $\mathcal{M}_l(\mathbf{x}; \boldsymbol{\theta}_l^p) - \mathcal{M}_s(\mathbf{x}; \boldsymbol{\theta}_s^p)$ on the training of $\mathcal{M}_s(; \boldsymbol{\theta}_s)$. Correspondingly, we introduce another coefficient $\alpha_{test}$ to Eqn. 2 during inference stage:

$$p(\mathbf{x}) = \mathbf{z}_{\mathcal{M}_s^t} + \alpha_{test}(\mathbf{z}_{\mathcal{M}_l^p} - \mathbf{z}_{\mathcal{M}_s^p}). \tag{4}$$

Especially, when $\alpha_{train} = \alpha_{test}$, our CPT maintains strict consistency by leveraging the consistent form of ensembling output logits from three models during both training and inference stages. While when $\alpha_{train} = 0$ in Eqn. 3 and $\alpha_{test} = 1$ in Eqn. 4, our CPT will degenerate into the "inconsistent" vanilla Proxy-tuning, *i.e.*, Eqn. 1 and Eqn. 2. In fact, Proxy-tuning can be considered

as a special case of our CPT. In Sec. 4.3, we will explore how the combinations of different $\alpha_{train}$ and $\alpha_{test}$ impact model performance. Note that our method focuses solely on the computation between the output logits of the model. Therefore, CPT can be widely applicable to various black-box model fine-tuning tasks which involve logit-level classification, such as image classification, image segmentation (pixel-level classification), text generation (in-vocabulary classification), etc. Furthermore, the large black-box model $\mathcal{M}_l$ and the smaller model $\mathcal{M}_s$ are not required to be from the same model family. They only require to share the same output space, e.g., the same classification categories in image classification tasks or the same vocabulary in text generation tasks. This allows our method to be flexibly applied to various combinations of black-box models and their white-box proxies.

### 3.3 Extending CPT to Vision-Language Model

To demonstrate that our method can be applied to other black-box model fine-tuning tasks involving logit-level classification, in this section we extend CPT to black-box Vision-Language Model (VLM) fine-tuning. VLMs have shown impressive capabilities across a diverse array of applications. Here, we mainly focus on applying CPT to black-box tuning for CLIP (Radford et al., 2021) on downstream image classification tasks. CLIP achieves image classification by calculating the similarity between image embeddings and different text embeddings. Formally, CLIP employs an image encoder $E_{img}(\cdot|\boldsymbol{\theta}_{img})$ and a text encoder $E_{txt}(\cdot|\boldsymbol{\theta}_{txt})$, which are jointly pretrained with a vast number of image-text pairs using a contrastive learning approach. Given an input image $\mathbf{x}$ and multiple tokenized descriptive texts $\mathbf{T} = \{\mathbf{t}_1, \mathbf{t}_2, \ldots, \mathbf{t}_C\}$ corresponding to $C$ classes, the image encoder and text encoder extract image embedding $\mathbf{f}$, and text embeddings $\{\mathbf{g}_c\}_{c=1}^C$ respectively, where $\mathbf{f} = E_{img}(\mathbf{x}|\boldsymbol{\theta}_{img})$ and $\mathbf{g}_c = E_{txt}(\mathbf{t}_c|\boldsymbol{\theta}_{txt})$. Then, the predicted logit score of class $c$ is computed by $\langle\|\mathbf{f}\|_2, \|\mathbf{g}_c\|_2\rangle$, where $\|\cdot\|_2$ denotes the $L_2$-normalization, and $\langle\cdot, \cdot\rangle$ denotes the cosine similarity of two embeddings. In the context of our CPT, we use a single symbol $M_*(\cdot|\boldsymbol{\theta}_*)$ ($*$ can be $s$ or $l$) to briefly represent both $E_{img}(\cdot|\boldsymbol{\theta}_{img})$ and $E_{txt}(\cdot|\boldsymbol{\theta}_{txt})$ of CLIP, where $\boldsymbol{\theta}_*$ represents the parameters of both $\boldsymbol{\theta}_{img}$ and $\boldsymbol{\theta}_{txt}$. We use $M_*(\mathbf{x}, \mathbf{T}|\boldsymbol{\theta}_*)$ to represent the classification logits predicted by the CLIP model for given $\mathbf{x}$ and $\mathbf{T}$. Therefore, the optimization objective of CPT for fine-tuning the black-box VLM can be expressed as:

$$\boldsymbol{\theta}_s^t = \arg\min_{\boldsymbol{\theta}_s} \mathbb{E}_{(\mathbf{x}, \mathbf{y}) \sim \mathcal{D}}[\mathcal{L}(\mathcal{M}_s(\mathbf{x}, \mathbf{T}; \boldsymbol{\theta}_s) + \alpha_{train}(\mathcal{M}_l(\mathbf{x}, \mathbf{T}; \boldsymbol{\theta}_l^p) - \mathcal{M}_s(\mathbf{x}, \mathbf{T}; \boldsymbol{\theta}_s^p)), \mathbf{y})]. \quad (5)$$

In practice, we use the CLIP with a heavy image encoder (*e.g.*, ViT-B/16 (Dosovitskiy et al., 2020)) as the large black box model, and an image encoder with a lighter image encoder (*e.g.*, ResNet-50 (He et al., 2016)) as the small white box model. During the training stage, we use templates like "a photo of a [CLS]", where [CLS] represents a certain class name, as inputs for the text encoder. Regarding the fine-tuning strategy of the small white-box model, we fine-tune all parameters of its image encoder and text encoder. In contrast to existing black-box fine-tuning methods for VLMs, which require access to image and text embeddings (Ouali et al., 2023; Guo et al., 2023), our method only requires access to cosine similarities (*i.e.*, output logits). This illustrates that our method can be applied to stricter black box model fine-tuning scenarios, where only logits are accessible.

## 4 Experiments

### 4.1 Experimental Setup

**Datasets.** For applying CPT to black-box tuning for LLM, we evaluate our CPT on TriviaQA (Joshi et al., 2017), ARC-challenge (Clark et al., 2018), commonsenseQA (Talmor et al., 2018), Corpus of Linguistic Acceptability (CoLA) (Warstadt et al., 2019), Microsoft Research Paraphrase Corpus (MRPC) (Dolan & Brockett, 2005), AG-News (Zhang et al., 2015) and Czech-to-English (Xu et al., 2023). For applying CPT to Black-box Tuning for VLM, we evaluate our CPT o CIFAR-10 (Krizhevsky et al., 2009), EuroSAT (Helber et al., 2019), Flowers102 (Nilsback & Zisserman, 2008), Stanford Cars (Krause et al., 2013), Oxford-IIIT Pets (Parkhi et al., 2012), Describable Textures Dataset (DTD) (Cimpoi et al., 2014), Country-211 (Radford et al., 2021) and Domainnet-10 (Peng et al., 2019). *Please refer to the supplemental materials sppl. B for more details.*

**Baselines.** For the experiments of both black-box tuning for LLM and VLM, we compare it with several other tuning settings to demonstrate the effectiveness of our proposed CPT: **a)** Zero-shot inference of pretrained black-box models on test set, which represents the baseline performance of

Table 1: **Comparison** of our CPT with other counterparts for black-box LLM tuning on seven natural language datasets. We treat LLAMA2-7B as the small white-box model and treat LLAMA2-13B as the large black-box model. "`pretrained`" represents the zero-shot inference by their official pretrained parameters. "`LORA-tuned`" represents directly fine-tuning the corresponding model with LORA. `Proxy-tuning` (Liu et al., 2024) and `CPT` represent using a 7B model to "proxy fine-tune" a 13B model, where the 7B model is trained using their method and our method, respectively. "ARC-C" and "cs2en" are the abbreviation of ARC-challenge and Czech-to-English.

| Model | Accuracy (%) ↑ | | | | | | | Mean Acc (%) ↑ |
|---|---|---|---|---|---|---|---|---|
| | TriviaQA | ARC-C. | commonsenseQA | COLA | MRPC | AG-News | cs2en. | |
| LLAMA2-7B | | | | | | | | |
| pretrained | 21.88 | 43.14 | 33.74 | 45.73 | 32.04 | 41.14 | 25.24 | 34.70 |
| LORA-tuned | 60.03 | 47.16 | 75.84 | 81.50 | 68.99 | 90.21 | 32.01 | 65.11 |
| LLAMA2-13B | | | | | | | | |
| pretrained | 36.76 | 53.85 | 35.71 | 70.95 | 67.96 | 64.15 | 33.19 | 51.80 |
| Proxy-tuning | 61.52 | 50.17 | 74.04 | 79.19 | 68.22 | 90.34 | 33.19 | 65.24 |
| **CPT (Ours)** | **62.79** | **55.85** | **76.41** | **82.26** | **69.77** | **90.91** | **34.07** | **67.44** |
| LORA-tuned | 66.58 | 66.22 | 81.90 | 84.65 | 68.99 | 90.65 | 35.54 | 70.64 |

Table 2: **Comparison** of our CPT with other counterparts for black-box VLM tuning on eight image classification datasets. We treat CLIP with ResNet-50 (RN-50) as the small white-box model and treat CLIP with ViT B/16 as the large black-box model. "`full-tuned`" represents directly fine-tuning the whole image encoder and text encoder of CLIP. `Proxy-tuning` (Liu et al., 2024) and `CPT` represent using a CLIP with RN-50 model to "proxy fine-tune" a CLIP with ViT B/16 model, where the CLIP with RN-50 model is trained using their method and our method, respectively. "DM-10.", "FL-102.", "CF-10.", "EUR.", "SC.", "OFP." and "CT211" are the abbreviation of Domainnet-10, Flowers102, CIFAR-10, EuroSAT, Stanford Cars, Oxford-IIIT Pets and Country-211, respectively.

| Model | Accuracy (%) ↑ | | | | | | | | Mean Acc (%) ↑ |
|---|---|---|---|---|---|---|---|---|---|
| | DM-10. | FL-102. | CF-10. | EUR. | SC. | OFP. | DTD | CT211. | |
| CLIP RN-50 | | | | | | | | | |
| pretrained | 81.15 | 66.09 | 70.37 | 36.16 | 53.94 | 85.69 | 40.27 | 14.18 | 55.98 |
| full-tuned | 88.83 | 74.78 | 94.13 | 98.44 | 74.43 | 87.38 | 65.32 | 20.20 | 75.43 |
| CLIP ViT-B/16 | | | | | | | | | |
| pretrained | 87.38 | 71.05 | 90.08 | 48.42 | 63.67 | 89.09 | 42.98 | 20.47 | 64.14 |
| Proxy-tuning | 90.14 | 76.96 | 95.21 | 98.33 | 76.93 | 89.10 | 65.69 | 25.07 | 77.17 |
| **CPT (Ours)** | **93.94** | **77.52** | **96.57** | **98.47** | **78.03** | **89.62** | **67.23** | **25.93** | **78.41** |
| full-tuned | 93.94 | 95.43 | 97.69 | 98.82 | 86.97 | 95.45 | 78.09 | 30.10 | 84.56 |

untuned black-box models. we compare our CPT with this untuned ones to show that our method can effectively perform tuning for black-box models. **b)** Fine-tuning the black-box models with Proxy-tuning (Liu et al., 2024). Proxy-tuning neglects the inconsistency between proxy-independent optimization during training and proxy-dependent probability distribution in inference stage, which results in sub-optimal solution. We compare CPT with Proxy-tuning to show that our model enhances performance by ensuring consistency between optimization objectives and inference-time proxy process. **c)** Fine-tuning the black-box model with white-box tuning methods. In fact, this tuning setting cannot be achieved in real-world scenarios for black-box model optimization due to the inability to access the internal parameters of the black-box model. We only use this setting as an ideal reference to assess how much our CPT lags behind direct fine-tuning in terms of performance. Additionally, we also compared zero-shot inference and direct tuning of small white-box models on each dataset.

**Implementation Details.** *Please refer to the supplemental materials sppl. C for more details.*

## 4.2 EXPERIMENTAL RESULTS

Tab. 1 and Tab. 2 shows the comparison of our CPT with other counterparts for black-box LLM tuning and for black-box VLM tuning respectively. We report the **Accuracy** for each dataset as well as the **Mean Accuracy** across all datasets.

Table 3: **Performance** of our CPT on models of different scale on MRPC (Dolan & Brockett, 2005) and ARC-challenge (Clark et al., 2018). In this particular case, a black-box LLAMA2-13B model is tuned with CPT with a white-box LLAMA-3B model as proxy.

| Model | Accuracy (%) ↑ | |
| --- | --- | --- |
| | MRPC | ARC-challenge |
| LLAMA-3B | | |
| pretrained | 52.97 | 23.41 |
| LORA-tuned | 68.22 | 33.11 |
| LLAMA2-13B | | |
| pretrained | 67.96 | 53.85 |
| Proxy-tuning | 67.96 | 52.84 |
| **CPT (Ours)** | **68.48** | **67.70** |
| LORA-tuned | 70.54 | 66.22 |

Table 4: **Comparison** of our CPT with other counterparts for black-box VLM tuning on Stanford Cars (Krause et al., 2013) and Oxford-IIIT Pets (Parkhi et al., 2012). The small white-box model, *i.e.*, CLIP RN-50, involved in Proxy-tuning and our CPT are tuned with CoOp (Zhou et al., 2022b).

| Model | Accuracy (%) ↑ | |
| --- | --- | --- |
| | Stanford Cars | Oxford-IIIT Pets |
| CLIP RN-50 | | |
| pretrained | 53.94 | 85.69 |
| CoOp-tuned | 77.83 | 91.20 |
| CLIP ViT-B/16 | | |
| pretrained | 63.67 | 89.09 |
| Proxy-tuning | 78.44 | 92.29 |
| **CPT (Ours)** | **81.66** | **93.21** |
| CoOp-tuned | 86.18 | 94.94 |

**Results of Black-box LLM Fine-tuning.** Tab. 1 shows the comparison results of black-box LLM fine-tuning on seven datasets. Clearly, our CPT significantly enhance the performance of pretrained model (*i.e.*, 13B `pretrained`) across all datasets. Moreover, CPT also outperforms Proxy-tuning (Liu et al., 2024) across all datasets, and surpass it by **2.20%**, in terms of Mean Accuracy, *i.e.*, 65.24% → **67.44%**. These results demonstrate that our CPT yields better fine-tuning effects on black-box LLMs compared to Proxy-tuning. We also observed that the performance of Proxy-tuning on several datasets are even worse than fine-tuning standalone white-box small models (*i.e.*, LLAMA-7B `LORA-tuned`). For instance, on commonsenseQA, COLA, and MRPC, the performance of Proxy-tuning are 1.80%, 2.31%, and 0.77% lower than that of 7B LORA-tuned, respectively. Note that the output of Proxy-tuned is an ensemble of outputs from a tuned small white-box model, a pretrained small white-box model, and a large black-box model. Therefore, it only makes sense if the performance of the large model being proxied exceeds that of the small model itself. However, the results of the ensemble by Proxy-tuning are worse than those of the single model, which implies that Proxy-tuning is still sub-optimized. In contrast, our method outperforms the 7B `LORA-tuned` across all datasets, also demonstrating that our CPT is superior to Proxy-tuning. From this perspective, our CPT can also serve as a novel fine-tuning method for white-box models. In this perspective, CPT seek guidance from a larger pretrained model to better fine-tune the smaller one. More interesting, on MRPC and Ag-News, our CPT even outperforms the method that hypothetically use white-box fine-tuning of large models (*i.e.*, 13B `LORA-tuned`). These results above fully demonstrate the effectiveness of our CPT for fine-tuning black-box LLM.

**Results of black-box VLM fine-tuning.** Tab. 2 shows the comparison results of black-box VLM fine-tuning on eight datasets. Similar to the results of black-box LLM fine-tuning, the standout aspects of our CPT can be summarized in the following four folds: **1)** Our CPT significantly improves the performance of pretrained VLM (*i.e.*, ViT-B/16 `pretrained`), and achieving **14.26%** improvement in terms of Mean Accuracy, *i.e.*, 64.14% → **78.41%** . **2)** Our CPT consistently outperforms Proxy-tuning (Liu et al., 2024) across all datasets, obtaining **1.23%** improvement in terms of Mean Accuracy, *i.e.*, 77.17% → **78.41%**. **3)** Proxy-tuning underperforms fine-tuning standalone white-box small models (*i.e.*, RN-50 `full-tuned`) on EuroSAT, *i.e.*, 98.44% → 98.33%, while CPT outperforms the RN-50 `full-tuned` across all datasets. **4)** Our CPT shows comparable performance with that of fine-tuning large models using white-box methods (ViT-B/16 `full-tuned`) on DomainNet-10, *i.e.*, 93.94% vs. 93.94%. To sum up, for both fine-tuning black-box LLM and VLM tasks, our CPT significantly improves the performance of pretrained large black-box model, and achieves higher performance compared to Proxy-tuning (Liu et al., 2024). This indicates that ensuring the consistency between training objectives and the formula of test-time proxy indeed further enhances the performance of the proxy-tuned model.

### 4.3 ABLATION STUDIES

**Using CoOp for White-box tuning in CPT.** Our CPT optimizes the large model on downstream tasks by fine-tuning a small white-box model as a proxy. In fact, CPT flexibly accommodates various fine-tuning strategies for the small white-box model. In the main paper, we adopt the fully fine-tuning for the image and text encoders of the small white-box model to act as a proxy (both Proxy-tuning (Liu et al., 2024) and CPT) for the black-box VLM model. Here, we use CoOp (Zhou et al., 2022b) to alternatively optimize the white-box model as an example to demonstrate CPT's in-

Table 5: **Performance** of our CPT on models of different scale on Stanford Cars (Krause et al., 2013), Oxford-IIIT Pets (Parkhi et al., 2012), DTD (Cimpoi et al., 2014) and Flowers102 (Nilsback & Zisserman, 2008). In this particular case, a black-box CLIP ViT-L/14 model is tuned with CPT with a white-box CLIP RN-50 model as proxy.

| Model | Accuracy (%) ↑ | | | |
|---|---|---|---|---|
| | SC. | OFP. | DTD. | FL-102. |
| CLIP RN-50 | | | | |
| pretrained | 53.94 | 85.69 | 40.27 | 66.09 |
| full-tuned | 74.48 | 87.35 | 65.32 | 74.78 |
| CLIP ViT-L/14 | | | | |
| pretrained | 76.74 | **93.49** | 52.93 | 77.54 |
| Proxy-tuning | 78.94 | 90.27 | 68.19 | 82.58 |
| **CPT (Ours)** | **81.69** | 91.58 | **69.31** | **84.75** |
| full-tuned | 91.92 | 97.00 | 82.07 | 97.35 |

Table 6: **Performance** of our CPT on models of different scale on Stanford Cars (Krause et al., 2013), Oxford-IIIT Pets (Parkhi et al., 2012), DTD (Cimpoi et al., 2014) and Flowers102 (Nilsback & Zisserman, 2008). In this particular case, a black-box CLIP ViT-L/14 model is tuned with CPT with a white-box CLIP ViT-B/16 model as proxy.

| Model | Accuracy (%) ↑ | | | |
|---|---|---|---|---|
| | SC. | OFP. | DTD. | FL-102. |
| CLIP ViT-B/16 | | | | |
| pretrained | 63.67 | 89.09 | 42.98 | 71.05 |
| full-tuned | 86.97 | 95.45 | 78.09 | 95.43 |
| CLIP ViT-L/14 | | | | |
| pretrained | 76.74 | 93.49 | 52.93 | 77.54 |
| Proxy-tuning | 88.67 | 96.18 | 78.09 | 96.02 |
| **CPT (Ours)** | **89.22** | **96.54** | **80.48** | **97.02** |
| full-tuned | 91.92 | 97.00 | 82.07 | 97.35 |

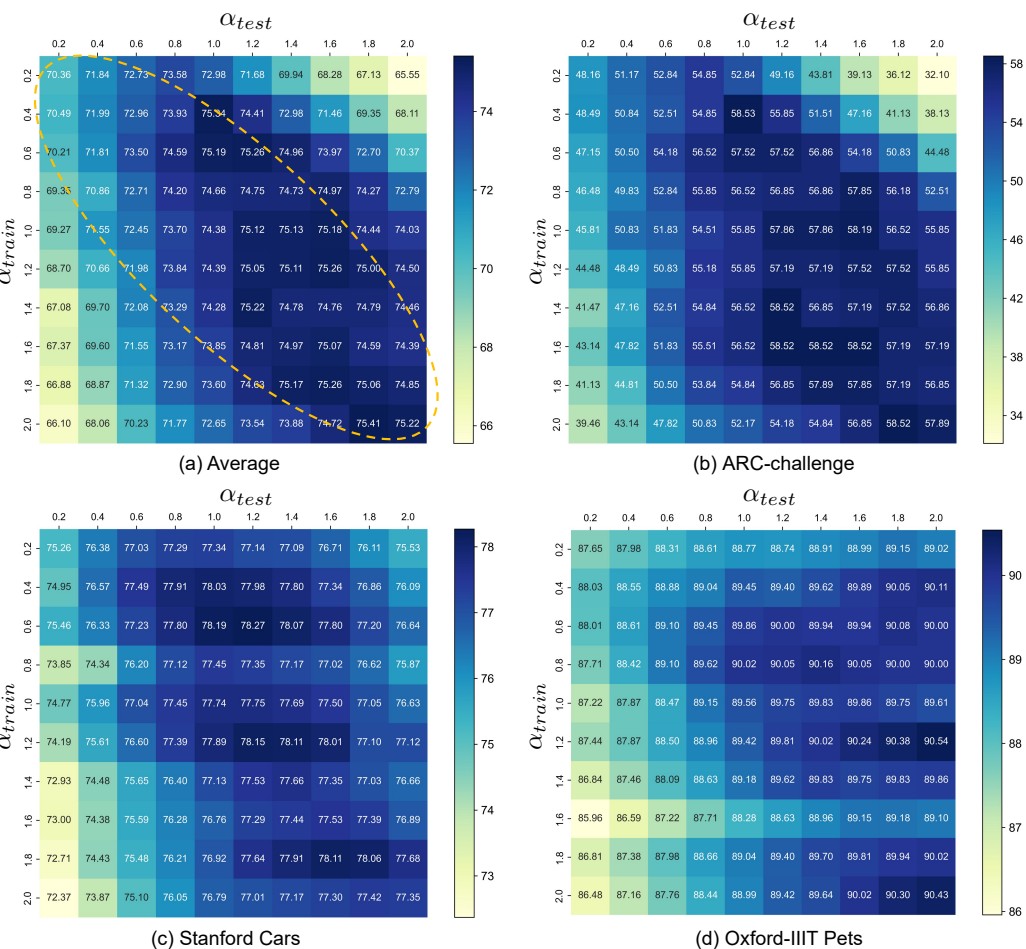

Figure 2: **Variation of the accuracy** versus the varied $\alpha_{train}$ and $\alpha_{test}$ on (b) ARC-challenge, (c) Stanford Cars, (d) Oxford-IIIT Pets and the results of their (a) Average .

clusivity towards the chosen white-box fine-tuning method. CoOp uses a set of learnable vectors to replace the text input template "a photo of a", which can efficiently fine-tune CLIP on downstream classification tasks. Tab. 4 shows the comparison results on Stanford Cars and Oxford-IIIT Pets. Note that different from Tab. 2, `Proxy-tuning` and `CPT` in Tab. 4 represent the use of a small white-box model fine-tuned with CoOp to act as a proxy for the black-box model. Similar to the re-

sults in Tab. 2, when using CoOp to fine-tune the small model, our CPT can still effectively enhance the performance of the large black-box model, and consistently outperforming Proxy-tuning.

**Tuning Black-box Models under Different Scales with CPT**   Here we demonstrate the effectiveness of our CPT method across various model architectures. In this case, we conduct experiments under various model architectures on tuning LLMs and VLMs. The main results shown in Tab. 1 is obtained with LLAMA2-7B as white-box proxy model and LLAMA2-13B as the large black-box model, and Tab. 2 is obtained with CLIP ResNet-50 as white-box proxy model and CLIP ViT-B/16 as the large black-box model. For tuning LLMs, we extend the experiment to tuning black-box LLAMA2-13B with CPT, LLAMA-3B being the white-box proxy. For tuning VLMs, we extend the experiment to tuning black-box CLIP ViT-L/14 with CPT, CLIP ResNet-50 and CLIP ViT-B/16 being white-box proxies respectively. The results in Tab. 3, Tab. 5 and Tab. 6 shows that CPT constantly outperforms Proxy-tuning and other baseline methods with different pairs of proxies and black-box models. Based on the experimental results, we infer that our method might also be effective in fine-tuning larger black-box models (e.g., LLAMA2-70B). However, due to limited computational resources, this part of the experiment will be left for future work.

**Effect of varied $\alpha_{train}$ and $\alpha_{test}$.**   $\alpha_{train}$ in Eqn. 3 and $\alpha_{test}$ in Eqn. 4 are hyper-parameters of our CPT. They determines the extent to which the offset calculated by $\mathcal{M}_l(\mathbf{x}; \boldsymbol{\theta}_l^p) - \mathcal{M}_s(\mathbf{x}; \boldsymbol{\theta}_s^p)$ affects the optimization objective and the proxy process. Intuitively, choosing a large $\alpha_{train}$ will amplify the impact of the difference between the outputs of the frozen large model and the small model on the optimization objective, while choosing a smaller one reduces this impact. Similarly, choosing different values of $\alpha_{test}$ will affect the final performance of the proxied model. Moreover, the relative values of $\alpha_{train}$ and $\alpha_{test}$ will affect the consistency between the training objective and the proxy process during testing. Specifically, the closer these two coefficients are, the stronger the consistency between the training objective and the proxy process during testing; conversely, the further apart they are, the weaker the consistency.

In all main experiments, both of these two coefficients are set to 1 to keep a strict consistency. Here, we further explore the impact of varied $\alpha_{train}$ and $\alpha_{test}$ on performance of CPT. Fig. 2 shows the performance of CPT under varied $\alpha_{train}$ and $\alpha_{test}$ on ARC-challenge, Stanford Cars (c), Oxford-IIIT Pets (d) and the results of their average (a). Each cell in this matrix represents the accuracy under a specific set of $\alpha_{train}$ and $\alpha_{test}$. From Fig. 2, we can obtain the following observations: **1)** The areas where the model performs well are concentrated near the main diagonal of the matrix, *e.g.*, marked with a yellow elliptical curve in Fig. 2 (a). When $\alpha_{train}$ and $\alpha_{test}$ are relatively close, the model tends to perform better; conversely, when they are further apart, the model's performance tends to decline. This result indicates that ensuring the consistency between $\alpha_{train}$ and $\alpha_{test}$ will be beneficial to the model's performance. This also indirectly supports our proposal in this paper for "ensuring the consistency between the optimization objectives and the proxy during testing can benefit for fine-tuning." In specific datasets, such as ARC-challenge and Stanford Cars, we can also observe similar conclusions. Due to the limitations of our hyperparameter selection range, this phenomenon is not observed in Fig. 2 (d). In Fig. 3 of supplemental material, we show the results under a wider range of hyperparameters, where the phenomenon is consistent with the other two datasets. **2)** From the results of each dataset, it can be seen that when $\alpha_{train}$ and $\alpha_{test}$ are close but both are relatively small, the performance of CPT is poor. This phenomenon may suggest that we should choose a relatively larger alpha when performing CPT. From the average results (2 (a)), when $\alpha_{train}$ and $\alpha_{test}$ are close and their values are between 1.2 and 2.0, the model performs well. We suggest choosing these two hyperparameters from the above range to perform CPT, for example, selecting $\alpha_{train} = \alpha_{test} = 1.2$.

## 4.4   INFERENCE & TRAINING COST

Here, we take finetuning LLMs as an example to analyze the time overhead of our CPT in both inference and training. Tab. 7 shows the comparison of our CPT with Proxy-tuning and single model in terms of inference time. We measure inference cost by the time (seconds) it takes to process each prompt-completion pair. Our analysis demonstrates that, like Proxy-tuning, our improved approach incurs no additional computational costs during inference. For the trainging time cost, to be honest, our method incurs higher costs than Proxy-tuning due to additional training-time predictions. We mitigate this by efficiently implementing a one-time inference model output for both small un-tuned and large black-box models at each step, storing them for future use. Tab. 8 shows the comparison of our CPT with Proxy-tuning and single model in terms of training time. "Extra cost" refers to

Table 7: Inference time cost of each compared method. Each value in the table represents the seconds taken to infer a single sample.

| Method | Inference time per sample | | | | | |
|---|---|---|---|---|---|---|
| | TriviaQA | ARC-C | commonsenseQA | COLA | MRPC | AG-News |
| LLAMA2-7B | 0.024 | 0.025 | 0.024 | 0.029 | 0.026 | 0.027 |
| LLAMA2-13B | 0.033 | 0.038 | 0.035 | 0.041 | 0.040 | 0.040 |
| Proxy-tuning (7B-to-13B) | 0.101 | 0.109 | 0.102 | 0.123 | 0.109 | 0.101 |
| CPT (7B-to-13B) | 0.101 | 0.109 | 0.102 | 0.123 | 0.109 | 0.101 |

Table 8: Training time cost of each compared method on MRPC. Each value in the table represents the minute taken to training on the whole dataset.

| Method | Basic time cost | Extra time cost | Total time cost |
|---|---|---|---|
| LLAMA2-7B-LORA | 124.5 | 0 | 124.5 |
| Proxy-tuning (7B-to-13B, LORA) | 124.5 | 0 | 124.5 |
| CPT (7B-to-13B, LORA) | 124.5 | 3.8 | 128.3 |

the joint inference process involving both models. We enhance accuracy by 1.55% on the MRPC dataset with a minimal 3.05% increase in training costs, training only for 2 epochs to maintain efficiency. Extending training would spread the "extra" costs across more epochs, reducing the cost per epoch, thus improving accuracy more significantly for less additional cost percentage-wise. For the VLM classification task, our efficient implementation slightly extends training time by minutes or seconds over several hours. This negligible extra cost for both training and inference is practically inconsequential.

## 5 LIMITATIONS

**Increase in computational resources.** While CPT does not necessarily increase computational expenses compared to directly tuning the black-box model during training, it surely increases computational expenses at inference stage. Facing the same problem with Proxy-tuning (Liu et al., 2024), *i.e.*, the time of inference increases because multiple models compute output logits jointly. Also, compared to the normal inference of a single black-box model, Proxy-tuning and CPT require more GPU memory to deploy the proxy models. Although we can implement CPT effectively by computing classification logits of pretrained models on train/test dataset and storing them for further use, the increase in computational expenses in inevitable during inference on new data.

## 6 BROADER IMPACTS

*Please refer to Sppl. D for more details.*

## 7 CONCLUSION

In this paper we proposed a simple yet effective black-box model tuning method named Consistent Proxy Tuning (**CPT**). We notice that vanilla Proxy-tuning (Liu et al., 2024) trains the white-box small model independently but uses an ensemble of the white-box and black-box models for inference. This inconsistency between the training objective and inference can lead to the proxy process being sub-optimal. In contrast, our CPT introduces the frozen black/white models into the fine-tuning process of the small model, thereby ensuring consistency between training-stage optimization objective and test-time proxy process. Our CPT can be plug-and-played for any black-box model fine-tuning tasks which involve logit-level classification. Extensive experiment results of the black-box tuning for VLMs and LLMs on many datasets demonstrate the effectiveness of our CPT.

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

# Supplemental Materials for CPT: Consistent Proxy Tuning for Black-box Optimization

## A DISCUSSION ABOUT EXPERIMENTS FOR INSTRUCTION-TUNING AND CODE-ADAPTATION

Proxy-tuning (Liu et al., 2024) employed their approach for Instruction-Tuning and Code-Adaptation tasks. In principle, we need to apply CPT to these two types of tasks and carry out a comparison to better demonstrate the effectiveness of our approach. However, Proxy-tuning used off-the-shelf models[1][2] for these tasks, utilizing models trained by others *without providing training data or details*. Lacking access to the original, proprietary training data and details, we *cannot replicate these models*. Therefore, we conducted our instruction-tuning using an alternative dataset[3], and the experiment results are shown in Tab. 9.

Table 9: Comparison of Instruction-Tuning for tuning LLM. Proxy-tuning does not work on selected dataset, making it impossible for us to work further on this basis.

| Model | Accuracy |
|---|---|
| LLAMA2 3b base | 21.78. |
| LLAMA2 3b lora | 35.37 |
| LLAMA2 13b base | 36.76 |
| LLAMA2 13b Proxy-tuned (Liu et al., 2024) | 29.46 |
| LLAMA2 13b lora | 90.14 |

We can see from above table that Proxy-tuning itself does not even work on this task/dataset, making it impossible for us to work further on this basis. However, we do extend our experiments to more new datasets and tasks (e.g. Arc-challenge, COLA, MRPC, AG-News, cs-to-en translation and coresponding tasks) which are not used in Proxy-tuning for a comprehensive analysis of our proposed method. A summary of our examined datasets/tasks is shown in Tab. 10.

Table 10: Summary of our examined datasets/tasks in this paper.

| Dataset | TriviaQA | ARC-C | commonsenseQA | COLA | MRPC | AG-News | cs-to-en |
|---|---|---|---|---|---|---|---|
| Task | QA-general | QA-choice | QA-choice | acceptability | paraphrase check | text classification | machine translation |
| Domain | general knowledge | natural science | general knowledge | linguistics | paraphrase examples | News | Czech-to-English linguistics |

## B MORE DETAILS OF DATASETS

**Datasets for LLMs.** We evaluate our CPT on TriviaQA (Joshi et al., 2017), ARC-challenge (Clark et al., 2018), commonsenseQA (Talmor et al., 2018), Corpus of Linguistic Acceptability (CoLA) (Warstadt et al., 2019), Microsoft Research Paraphrase Corpus (MRPC) (Dolan & Brockett, 2005), AG-News (Zhang et al., 2015) and Czech-to-English translation subset of ALMA-Human-Parallel (Xu et al., 2023). These datasets cover common natural language understanding tasks, including question-answering, linguistic analysis, paraphrasing and translation. Note that some datasets are often formulated as text classification tasks, for example, adding a classification header to the last layer of the model to predict the result. However, this is not feasible for black-box LLMs. Therefore, we convert all tasks into text generation tasks for processing. Specifically, for each dataset, we construct specific prompts to standardize the output format of the model as much as possible. We calculate the accuracy of the model by matching the text generated by the model with the ground truth labels. Tab. 11 shows the details of each dataset, including train size, test

---

[1]https://huggingface.co/meta-llama/Llama-2-7b-chat-hf
[2]https://github.com/Meta-Llama/codellama
[3]https://github.com/yizhongw/self-instruct/blob/main/data/gpt3_generations/batch_221203/all_instances_82K.jsonl

size and used prompts. During inference stage, {question} will be filled with a specific question of one data point, combining with contextual template text to form a complete prompt as input to the model. While during training stage, the answers will also be filled into the {prediction} and combined with the previous question parts to form the input for training the model. All datasets for LLMs are in the default version on their offical website. Licenses for datasets are also specified in Tab. 11. N/A indicates that there is no explicit license on the official website and the we are reaching out to original authors of the assets.

**Datasets for VLMs.** For applying CPT to Black-box Tuning for VLM, we first choose 7 well-studied image classification datasets which cover a variety of data distributions, *i.e.*, CIFAR-10 (Krizhevsky et al., 2009), EuroSAT (Helber et al., 2019), Flowers102 (Nilsback & Zisserman, 2008), Stanford Cars (Krause et al., 2013), Oxford-IIIT Pets (Parkhi et al., 2012), Describable Textures Dataset (DTD) (Cimpoi et al., 2014), and Country-211 (Radford et al., 2021). Followed Li *et al.* (Li et al., 2021), we also evaluate our CPT on one more challenging datasets with inner domain gap, *i.e.*, Domainnet-10 (Peng et al., 2019). Domainnet-10 contains top-10 classes based on data amount in DomainNet which has 345 categories. For each dataset, we use specific prompts to the mentioned in Sec. 3.3 to better adapt the model to domain-specific knowledge. And we utilize the official category names provided in corresponding datasets to complete the template prompt. Tab. 12 shows the details of each dataset, including train size, test size and used prompts. All datasets for VLMs are used in the torchvision version, except for Domiannet-10 (Li et al., 2021) dataset and EuroSAT (Helber et al., 2019) dataset, which are in the same version with FedBN (Li et al., 2021). Licenses for datasets are also specified in Tab. 12. N/A indicates that there is no explicit license on the official website and the we are reaching out to original authors of the assets.

## C   IMPLEMENTATION DETAILS

All experiments are conducted with PyTorch toolkit (Paszke et al., 2019) on NVIDIA A100-40G GPU. For black-box tuning of LLM, we use LLAMA2-7B as the small white-box model (*i.e.*, $\mathcal{M}_s$), and use LLAMA2-13B as the large black-box model (*i.e.*, $\mathcal{M}_l$). For the training stage, we adopt LoRA to fine-tune the small white-box model $\mathcal{M}_s(\cdot; \boldsymbol{\theta}_s^p)$ for computational efficiency. Note that the large black-box model $\theta_l^p$ and another small white-box $\theta_s^p$ are only responsible for providing output logits for optimization objectives, and their own parameters are frozen throughout the entire training stage. For training configurations, AdamW is used as the optimizer, with a initial learning rate of 1e-4, batch size is set to 1 and model is trained for 2 epochs. For black-box tuning of VLM, we use CLIP with ResNet-50 as the small white-box model (*i.e.*, $\mathcal{M}_s$), and use CLIP with ViT-B/16 as the large black-box model (*i.e.*, $\mathcal{M}_l$). For the training stage, we fully fine-tune the whole image encoder and text encoder of the white-box model $\mathcal{M}_s(\cdot; \boldsymbol{\theta}_s^p)$. For training configurations, Adam is used as the optimizer, with a momentum of 0.9 and weight decay of 0.001, batch size is set to 128 and model is trained for 300 epochs. For all the experiments, we set $\alpha_{train} = \alpha_{test} = 1$ by default.

## D   BROADER IMPACTS

**Positive Impact: Fairness in AI.** CPT aims to tune a black-box model with smaller "proxies" consistently, but it can also serve as a fine-tuning method for smaller models guided by large pretrained models. Large-scale pretraining is more often on general knowledge than domain-specific knowledge, so downstream tasks barely benefit from large-scale pretrained models without computationally expensive fine-tuning. It is even less possible when large pretrained model is only accessible as black-boxes. CPT fills the gap by joining the strength of large-scale general knowledge pretraining and small-scale task-specific fine-tuning. From this perspective, CPT brings positive social impact in that it finds a way of using general pretrain model to elevate task-specific fine-tuning of smaller models. It is of great significance for individuals, organizations and regions without resources to fine-tune large pretrain models for their own well-being, thus improving fairness of AI.

**Negative Impact: Potential Misuse.** Black-box language models, compared with white-box ones, are more difficult to tune for specific tasks. While this limits the application of black-box language models, it also prevents them from being misused to generate malignant content. However, the way

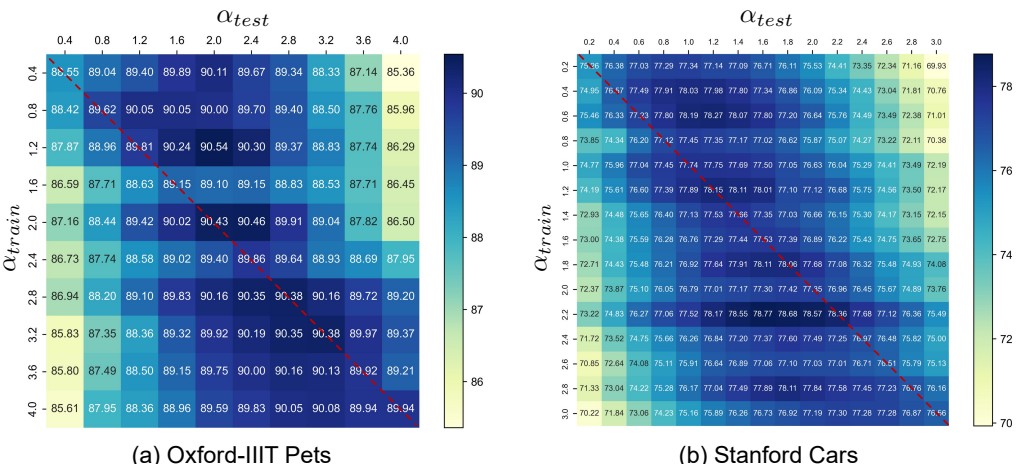

(a) Oxford-IIIT Pets            (b) Stanford Cars

Figure 3: **Variation of the accuracy** versus the widely varied $\alpha_{train}$ and $\alpha_{test}$ on (a) Oxford-IIIT Pets (b) and Stanford Cars .

CPT tunes a black-box model formulates a by-pass around inaccessible parameters to output logits, making the output logits as volunrable to harmful content as it is adjustable.

Harm control methods like gated release of models, API monitoring for misuse, limitation on access frequency to prevent API-based CPT should be considered to minimize negative social impacts.

# E    EXTENDED RESULTS

In addition to the partial results shown in Fig. 2, we demonstrate the extensive results of CPT on VLMs for Oxford-IIIT Pets dataset and Stanford Cars dataset. To demonstrate the impact of "consistency", *i.e.*the extent to which $\alpha_{train}$ and $\alpha_{test}$ are close to each other. To better demonstrate the whole pattern, we sample $\alpha_{train}$ and $\alpha_{test}$ from 0.4 to 4 with an inverval of 0.4 for Oxford-IIIT Pets dataset, and $\alpha_{train}$ and $\alpha_{test}$ from 0.2 to 3.0 with an interval of 0.2 to 3.0 for Stanford Cars dataset. Both matrices of results in Fig. 3 are obtained with the same setup with results in Tab. 2 except for $\alpha_{train}$ and $\alpha_{test}$. The extended results demonstrate the same pattern with that in Sec. 4, *i.e.*, the performance of the models are better when $\alpha_{train}$ and $\alpha_{test}$ are closer and formulate a "consistency".

Table 11: Details of each dataset of fine-tuning black-box LLM task.

| Dataset | Train size | Test size | Prompt | License |
|---|---|---|---|---|
| TriviaQA (Joshi et al., 2017) | 87,622 | 11,313 | Question: {question}

Answer: {prediction} | Apache 2.0 |
| ARC-challenge (Clark et al., 2018) | 1,119 | 299 | Question: {question}
Please choose:
A. {option A}
B. {option B}
C. {option C}
D. {option D}

Answer: {prediction} | Apache 2.0 |
| commonsenseQA (Talmor et al., 2018) | 9,741 | 1,221 | Question: {question}
Please choose:
A. {option A}
B. {option B}
C. {option C}
D. {option D}
E. {option E}

Answer: {prediction} | N/A |
| COLA (Warstadt et al., 2019) | 8,551 | 1,043 | Sentence: {sentence}
Question: Is this sentence linguistically acceptable? (Yes or No)

Answer: {prediction} | N/A |
| MRPC (Dolan & Brockett, 2005) | 3,527 | 387 | Sentence 1: {sentence 1}
Sentence 2: {sentence 2}
Question: Are these two sentences expressing the same meaning? (Yes or No)

Answer: {prediction} | N/A |
| AG-News (Zhang et al., 2015) | 120,000 | 7,599 | Given the following news article: {sentence}
Question: what category does this article belong to? Please choice:
A. {option A}
B. {option B}
C. {option C}
D. {option D}

Answer: {prediction} | N/A |
| ALMA-Human-Parallel (Xu et al., 2023) | 121,000 | 1000 | "cs": "Osmadvacetiletý šéfkuchař nalezen mrtev v obchodě v San Francisku",
"en": "28-Year-Old Chef Found Dead at San Francisco Mall" | N/A |

Table 12: Details of each dataset of fine-tuning black-box VLM task.

| Dataset | Train size | Test size | Prompt | License |
|---|---|---|---|---|
| DomainNet-10 (Peng et al., 2019) | 18,278 | 4,573 | A photo of a [CLASS] . | N/A |
| Flowers-102 (Nilsback & Zisserman, 2008) | 1,020 | 6,149 | A photo of a [CLASS] , a type of flower. | N/A |
| CIFAR-10 (Krizhevsky et al., 2009) | 50,000 | 10,000 | A photo of a [CLASS] . | N/A |
| EuroSAT (Helber et al., 2019) | 13,500 | 8,100 | A centered satellite photo of [CLASS] . | MIT License |
| Stanford Cars (Krause et al., 2013) | 8,144 | 8,041 | A photo of a [CLASS] . | N/A |
| Oxford TIII Pets (Parkhi et al., 2012) | 3,680 | 3,669 | A photo of [CLASS] , a type of pet. | CC BY-SA 4.0 |
| DTD (Cimpoi et al., 2014) | 1,880 | 1,880 | A photo of a [CLASS] texture. | N/A |
| Country-211 (Radford et al., 2021) | 31,650 | 21,100 | A photo I took in [CLASS] . | MIT License |

