# OpenReview forum: "CPT: Consistent Proxy Tuning for Black-box Optimization"
_ICLR.cc/2025/Conference — ICLR 2025 Conference Withdrawn Submission_

### Official Review · Reviewer_YQTs · 2024-11-02

**Soundness:** 2
**Presentation:** 3
**Contribution:** 1
**Rating:** 3
**Confidence:** 4

**Summary:**

The paper considers the setting of black-box tuning of models. The authors build upon recently proposed Proxy Tuning method [1] and introduce the usage of black-box frozen model into the training objective to make the training and the inference stage consistent, that, in-turn leads to slightly improved performance upon the original method. The authors evaluate the proposed approach on vision tasks using CLIP-like models and on NLP tasks with Llama family of models.


[1] Liu et al. Tuning language models by proxy. COLM 2024

**Strengths:**

* The considered setting of black-box tuning is important and is of interest of a broad community given that currently the most capable models are close-weight models.

**Weaknesses:**

* The methodological contribution is the very minor addition to the original method [1]. The obtained quantitive results also demonstrate minor improvements upon the original Proxy Tuning.

* Big part of the experiments for vision tasks is done for CLIP-like models. The considered setting of black-box fine-tuning of CLIP-like models is not well-justified since to the best of my knowledge there are no closed-source CLIP-like VLMs. I strongly encourage authors to consider the setting that is closer to the original paper and evaluate the proposed methodology on autoregressive VLMs like OpenFlamingo [2], or Llama 3.2 Vision [3]. For such class of models there are indeed closed-source models like GPT-4o and the setting would make more sense.

[1] Liu et al. Tuning language models by proxy. COLM 2024

[2] Awadala et al. OpenFlamingo: An Open-Source Framework for Training Large Autoregressive Vision-Language Models. arXiv:2308.01390

[3] Dubey et al. The Llama 3 Herd of Models. arXiv:2407.21783

**Questions:**

The considered setting aims to improve the predictions of the black-box model. Prompting techniques such as CoT [1] and others can also be applied in such black-box setting, thus, I encourage authors to include such baselines at least for NLP tasks to strengthen the paper.

[1] Kojima et al. Large Language Models are Zero-Shot Reasoners. NeurIPS 2022.

---

### Official Review · Reviewer_U5LU · 2024-11-04

**Soundness:** 2
**Presentation:** 2
**Contribution:** 1
**Rating:** 3
**Confidence:** 3

**Summary:**

This paper addresses the black-box optimization problem for language models and vision-language models. It proposes consistent proxy tuning to directly optimize the small proxy model in proxy-tuning [1] jointly with the target black-box model. In contrast to the vanilla proxy-tuning which optimizes the small proxy model alone, the joint optimization improves the accuracy of proxy-tuning while it introduces additional complexities explained below.

[1] Liu et al., "Tuning language models by proxy." (2024)

**Strengths:**

1. The paper is easy to follow.
2. The proposed method is straightforward and easy to implement.
3. Extensive experiments are performed with language models and vision-language models. Actual training time is also reported.

**Weaknesses:**

1. The novelty of the proposed method is limited. It just directly optimizes the small proxy model in proxy-tuning to achieve more accuracy, which is a trivial remedy and even hurts some advantages of proxy-tuning, i.e., it introduces additional memory/computational cost and requires the availability/accessibility to the black-box model during optimization.
2. The gains of consistent proxy tuning in accuracy seem too little against the vanilla proxy-tuning in most cases, relative to zero-shot and white-box tuned results, even though the proposed method exploits the (specific) black-box target model during optimization.  In such cases, the advantage by the consistent proxy tuning may not outweigh its disadvantages like additional cost and the need of availability of the black-box model.
3. The paper ignores a large portion of previous works [2-8] in black-box optimization for language models or vision-language models and avoids experimental comparison to them. Particularly, Ormazabal et al [2] proposed a very similar method that tunes the small proxy model and then combines it with the output of a target black-box model. The previous method in this field should be compared with the proposed method, not only in accuracy but also in the number of API calls for black-box optimization.

[2] Ormazabal et al., "CombLM: Adapting Black-Box Language Models through Small Fine-Tuned Models." (2023)

[3] Diao et al., "Black-box Prompt Learning for Pre-trained Language Models" (2022)

[4] Cheng et al., "Black-Box Prompt Optimization: Aligning Large Language Models without Model Training." (2023)

[5] Chen et al., "InstructZero: Efficient Instruction Optimization for Black-Box Large Language Models" (2023)

[6] Guo et al., "Connecting Large Language Models with Evolutionary Algorithms Yields Powerful Prompt Optimizers." (2023)

[7] Oh et al., "BlackVIP: Black-Box Visual Prompting for Robust Transfer Learning." (2023)

[8] Liu et al., "Language Models as Black-Box Optimizers for Vision-Language Models." (2023)

**Questions:**

See weaknesses.

---

### Official Review · Reviewer_kFZ9 · 2024-11-04

**Soundness:** 3
**Presentation:** 3
**Contribution:** 3
**Rating:** 5
**Confidence:** 3

**Summary:**

The paper introduces a method for fine-tuning black-box models called Consistent Proxy Tuning (CPT). This approach addresses the limitations of existing methods, such as Proxy-tuning, by ensuring consistency between the training objective and the inference process. CPT achieves this by incorporating the same regularization of the large model's logits in both the training and inference stages.

**Strengths:**

1. The paper is well-written, with clear logic that is easy to follow.

---
2. The proposed method is clear, well-motivated, and demonstrates simplicity while being effective.

---
3. The method uses the output logits of the large model as a reference. By incorporating these logits into the loss function of the small model, the training objective becomes closer to the ensemble effect used during inference. This acts as an implicit regularization on the small model during training, ensuring its output aligns with the actual inference combination form.

---
4. The experiments are good, covering both LLMs and VLMs to demonstrate the method's effectiveness across different model types.

**Weaknesses:**

1. Although the method is effective, it requires significant computational resources, especially during training when additional model outputs are generated. This could limit its practicality for larger-scale models without high-resource infrastructure.

---
2. The impact of parameters like $\alpha_{\text{train}} $ and $ \alpha_{\text{test}}$ on performance, shown in Figure 2, indicates that careful tuning is necessary, which may complicate implementation.

---
3. The motivation states, “Such inconsistency may lead to sub-optimal solutions for the proxy tuning optimization objective, causing bottlenecks in model performance.” While this is intuitive and understandable, it would be better if the paper provided deeper analysis or references to relevant literature that demonstrate or explain why consistency is beneficial in such contexts.

---
I appreciate the contributions of this paper, and if my question could be addressed, I would be open to considering a higher score.

**Questions:**

I am wondering if maintaining consistency between training and testing/inference is always beneficial. For instance, many preference optimization algorithms, such as Direct Preference Optimization (DPO), do not maintain complete consistency between training and testing. I am curious about the scenarios where such consistency is advantageous and why, in some cases, consistency is not a primary focus.

---

### Official Review · Reviewer_V8SH · 2024-11-04

**Soundness:** 3
**Presentation:** 3
**Contribution:** 3
**Rating:** 5
**Confidence:** 4

**Summary:**

Modern foundation models, unlike traditional ML models, usually do not provide oracle access to their feature spaces and gradients.  With access restricted to logits, proxy-tuning thus emerges as a promising tool to "fake" tune the base model by tuning a smaller model and shift the output change between original and tuned small models onto the base model. Traditional proxy tuning decouple training and inference into two different stages where a small model is first fine-tuned on its own and the base model is fake-tuned during inference only. However, this creates an objective gap between the small model and large model.

To bridge this gap, this paper presents a simple online tuning method by tuning the small model while distilling from the base model's logits at the same time. Experiments show the effectiveness of the method.

**Strengths:**

1. The proposed method is elegant, simple yet effective and beneficial to practitioners.
2. The presentation is clean and easy to follow.
3. The proposed method is extended to VL models to further demonstrate its effectiveness.

**Weaknesses:**

1. Lack of analysis over why proxy tuning works: This is a big ask, but I hope the authors can give some intuitions here. In addition, if the underlying assumption is that both small and base model can access the output logits for the entire vocabulary, then how is the inconsistency defined. Can you elaborate more? How about in CLIP's case.
2. Increased actual cost: Although not requiring more computation during training, there is a need to compute logits using the black box model for each training data. If we are looking at true black box models, as the authors claimed, such as GPT, Gemini, then we are looking at some potential big bill $$.
3. Questionable experimental designs: the authors seem to position their work as a type of PEFT for generic fine-tuning which I found somewhat problematic since LORA FT requires significantly fewer parameters than tuning even the base model here. I think the authors should position their work similarly to [1] as task-oriented efficient fine-tuning and find more interesting cases to demonstrate the effectiveness (domain adaptation, imbalanced learning, to name a few).

[1] Liu et. al., Tuning Language Models by Proxy

**Questions:**

See weakness above. While I appreciate the simplicity of the proposed method, my current verdict places it just under the acceptance bar due to increased cost and somewhat questionable experimental designs. I look forward to the authors' responses.

**Details Of Ethics Concerns:**

No ethics concerns are found

---

### Note · Authors · 2024-11-23

I have read and agree with the venue's withdrawal policy on behalf of myself and my co-authors.